# Estimating the Feeding Time of Individual Broilers via Convolutional Neural Network and Image Processing

**DOI:** 10.3390/ani13152428

**Published:** 2023-07-27

**Authors:** Amin Nasiri, Ahmad Amirivojdan, Yang Zhao, Hao Gan

**Affiliations:** 1Department of Biosystems Engineering and Soil Science, University of Tennessee, Knoxville, TN 37996, USA; anasiri@utk.edu (A.N.); aamirivo@vols.utk.edu (A.A.); 2Department of Animal Science, University of Tennessee, Knoxville, TN 37996, USA; yzhao@utk.edu

**Keywords:** broiler, feeding time, YOLO, image processing

## Abstract

**Simple Summary:**

Using automated approaches to investigate feeding behavior in broilers provides accurate, non-invasive, and large-scale data collection, real-time monitoring capabilities, and opportunities for advanced data analysis that would not be possible with manual observations. These benefits contribute to a better understanding of broilers’ behavior for improving production efficiency and animal welfare, optimizing management practices, and promoting the profitability of poultry production. Hence, this study aimed to estimate the feeding time of individual broilers through an automated approach. First, the proposed algorithm detected the broilers’ heads. Then, a Euclidean distance-based tracking algorithm tracked the detected heads. The developed algorithm can estimate the broiler’s feeding time by identifying whether its head is inside the feeder area. The overall accuracy of each broiler’s feeding time per visit to the feeding pan was 87.3%.

**Abstract:**

Feeding behavior is one of the critical welfare indicators of broilers. Hence, understanding feeding behavior can provide important information regarding the usage of poultry resources and insights into farm management. Monitoring poultry behaviors is typically performed based on visual human observation. Despite the successful applications of this method, its implementation in large poultry farms takes time and effort. Thus, there is a need for automated approaches to overcome these challenges. Consequently, this study aimed to evaluate the feeding time of individual broilers by a convolutional neural network-based model. To achieve the goal of this research, 1500 images collected from a poultry farm were labeled for training the You Only Look Once (YOLO) model to detect the broilers’ heads. A Euclidean distance-based tracking algorithm was developed to track the detected heads, as well. The developed algorithm estimated the broiler’s feeding time by recognizing whether its head is inside the feeder. Three 1-min labeled videos were applied to evaluate the proposed algorithm’s performance. The algorithm achieved an overall feeding time estimation accuracy of each broiler per visit to the feeding pan of 87.3%. In addition, the obtained results prove that the proposed algorithm can be used as a real-time tool in poultry farms.

## 1. Introduction

Monitoring broiler behaviors such as feeding, drinking, and perching is a crucial aspect of precision livestock farming to reflect their health status and provide early disease warning. In this regard, the feeding behavior of broilers plays a critical function in the breeding process. Deviation from the daily food consumption pattern is the early disease symptom. Thus, understanding poultry feeding behavior helps evaluate their use of feed resources, improve their health status, and provide vital economic and welfare implications for poultry production. As a result, new techniques are needed to extract the feeding behavior of broilers that are useful in warning about their health status and improving the breeding process.

Several studies have been conducted to investigate the feeding behavior of poultry under the influence of various environmental stimuli, management practices, and breeding systems. These studies often monitor poultry behaviors by manual observation or remotely [1]. Manual observation is an accurate and simple method to analyze the behavior of small samples and limited behavioral responses. However, manual observation is time-consuming and laborious, especially for large farms or monitoring multiple behaviors simultaneously. Therefore, it is necessary to develop automated methods to handle large sample sizes and multiple behaviors.

Broilers’ behavior on the experimental or commercial farm can be examined by applying automated techniques (e.g., audio analysis, radio-frequency identification (RFID) devices, and image processing) to analyze the health and welfare of the poultry. For example, audio analysis can be applied as a poultry behavior-based early warning system, detecting growth rates and predicting health conditions. Collecting and analyzing the individual chickens’ sounds can distinguish them from each other. In this respect, detecting which broiler makes the sound, collecting individual sounds on a farm, and ambient noise are significant challenges [2]. Wireless wearable sensors such as accelerometers and RFID microchips are primarily used to track people’s location and activity remotely. The performance of the RFID system depends on the number of broilers and installed antennas. Due to the large number of broilers on commercial farms and the sensor cost in relation to the value of the individual bird, attaching an RFID device to every broiler is currently unrealistic. In addition, RFID systems can be employed for a limited number of broilers due to the time-consuming tasks of installing and recycling tags [2].

The image processing technique is an efficient, non-invasive, and cost-effective method for analyzing animal behavior. This technology includes image-capturing systems and various algorithms to recognize the behavior. Kashiha et al. [3] investigated real-time broiler distribution indicators using image-processing methods. These authors reported unusual feeding and drinking behaviors in broilers with 95% accuracy. Also, Nasiri et al. [4] proposed a computer vision-based system to recognize lameness in broilers. Despite using image processing techniques to analyze particular poultry behaviors, there needs to be more research on their application in investigating broiler feeding behaviors.

As a specialized version of image processing, video monitoring is a low-cost and straightforward method to detect feeding behavior. In this regard, effectively extracting information from surveillance videos is a fundamental problem. For each frame, the first step is to distinguish the broilers from the background. Segmentation thresholding based on histogram analysis, Otsu segmentation, the maximum entropy segmentation method, and multi-level threshold segmentation are among the methods applied to detect objects from the background [3,5,6]. Their performance depends on the differentiation between broilers from various background conditions and varying light conditions. Furthermore, it is difficult to distinguish them individually when broilers are huddled together. Each broiler can be detected in the video sequence by creating specific marks on every broiler and using pattern recognition techniques. Therefore, the automatic recognition of broilers through video footages is a fundamental issue, and selecting the appropriate feature is crucial.

The performance of image processing technology can be improved using machine learning methods. Valletta et al. [7] utilized PCA to extract pheasant egg characteristics and k-means clustering to identify individual pheasant eggs. Kasinathan et al. [8] used shape features extracted from various categories of insect images and machine learning algorithms, including artificial neural networks, support vector machine, k-nearest neighbors, naive Bayes, and convolutional neural networks, to classify the insect classes. Another study proposed an image processing and machine learning framework for leaf disease detection. In this framework, the k-means and principal component analysis approaches were applied to segment and extract the features from leaf images to evaluate the disease. Then, images were classified using RBF-SVM, SVM, random forest, and ID3 techniques [9]. Bai et al. [10] introduced a vision-based algorithm for picking point localization of tomatoes. The proposed algorithm extracted the shape, texture, and color features and applied the SVM classifier to recognize tomatoes. In this regard, deep learning algorithms, as a type of machine learning method, have been used to identify and classify objects in many applications, including animal identification and behavior recognition. For example, Chen et al. [11] proposed a method to define the sex of pigeons by enhancing images using image processing techniques combined with YOLO-v5. In another study, the YOLO-v5 structure-based method was developed to monitor cage-free hens’ spatial distribution, including the number of birds in the perching, feeding, drinking, and nesting zones [12]. Li et al. [13] developed a Faster R-CNN-based algorithm to detect and track birds walking around the feeder as a feeding behavior indication. Guo et al. [14] evaluated the performance of different deep learning models such as ResNet, EfficientNet, and DenseNet to identify four broiler behaviors: (1) resting, (2) standing, (3) feeding, and (4) drinking. In their study, the accuracy of the best network for broiler behavior classification was 97%.

Monitoring broilers’ feeding behaviors can help ensure an appropriate diet to support their growth. Additionally, observing feeding behaviors can provide insights into the health and welfare of broilers. For instance, any changes in appetite may reflect underlying health issues. As a result, the feeding assessment of individual broilers assists in implementing proper nutritional management strategies. Hence, monitoring the feeding behaviors of individual broilers is crucial for precision livestock farming, intending to ensure optimal growth and maintain health and welfare. Accordingly, the objective of the study was to develop an algorithm based on image processing techniques and a deep learning model with the aim of recognizing the individual feeding times of broilers. To the extent of the authors’ knowledge, the present study was one of the first efforts to investigate the individual broilers’ feeding time on a commercial farm. The performance of the developed algorithm was validated by comparing the achieved results with the manual observations in the labeled videos.

## 2. Materials and Methods

### 2.1. Acquisition of Broilers’ Video

Data were collected at a commercial-scale broiler research farm (Tyson Foods, Inc., Huntsville, AR, USA) with 20,000 birds and a stocking density of 12.2 birds/m^2^. A total of 12 surveillance cameras (Dahua Technology USA Inc., Irvine, CA, USA) were installed on the ceiling of the farm (approximately 3 m above ground) to collect RGB videos with a set speed of 15 frames per second (fps). The cameras were fixed and distributed uniformly along the two drinker lines on the farm. In addition, four feeder pans were also in the cameras’ field of view. Videos were collected 24/7 for the entire cycle of several flocks in 2022.

### 2.2. Head Detection Model and Data Collection

Animals’ feeding behavior can be chewing, biting, or putting the head in the feeder (named a feeding visit). Since it is difficult to tell whether broilers are chewing or biting, a feeding visit can be commonly decided by checking whether the broiler’s head is in the feeder. Accordingly, this study defines broiler feeding behavior as broilers placing their heads in the feeder area. In this process, the feeder occupation time by the broiler’s head is calculated as feeding time. Therefore, for each frame, it is obligatory to detect the broiler’s head as a factor closely related to the detection of feeding behavior. In this study, a regression-based algorithm was used to address the broilers’ head detection issue.

You Only Look Once (YOLO), proposed in 2016, formulates the object detection problem as a regression problem [15]. Compared to two-stage detectors, YOLO is speedy. YOLO calculates the region of interest and image classes in one algorithm implementation. First, a neural network is processed on the whole image. Then, the image is divided into different cells, and the objects in each cell are projected [16,17]. YOLO divides the input image into a grid and predicts bounding boxes and class probabilities directly from the grid cells. This approach makes YOLO faster and more efficient. Each cell predicts objects that have their centers within that cell. On the other hand, YOLO predicts multiple bounding boxes for each grid cell using a set of anchor boxes. In recent years, researchers have been utilizing YOLO to identify individual animals and their behavioral patterns, which can result in a better investigation of animals’ behavioral mechanisms [18,19,20,21,22].

The database used in this study was created by selecting sample frames from surveillance video sequences to train and test the network. A more diverse database can be achieved by selecting frames with different postures of broilers (e.g., standing, lying, sitting, and with different lighting conditions). A total of 1500 images were selected for manual labeling. Figure 1 demonstrates an example of the labeled image. Then, the head of each broiler was labeled in the text format expected for training the YOLO-v3 model. In this study, the transfer learning method was adopted to solve the problem of the insufficient number of samples during the training process. For training the pre-trained YOLO on the COCO dataset, the dataset images were randomly divided into training and validation subsets with a ratio of 9:1.

### 2.3. Broiler’s Head Tracking Algorithm

YOLO object detection does not treat objects in every video frame the same. In other words, each broiler detected in the video frames is assigned a unique tag/identifier. Thus, it is necessary to use the tracking algorithm to assign a constant identifier for each detected broiler and for it to have high efficiency in the later stages. In each frame, the broiler tracking algorithm calculates the central point of the bounding box around the broiler’s head marked by the trained YOLO model. Then, the algorithm delivers the central point along with the specific identifier to an array that stores the characteristics of the detected broilers in the last 10 frames. In the next step, the Euclidean distance between the coordinates of the central point of the broiler’s head identified in the current frame is measured with the coordinates of the center points of the previous frame. The broiler receives its last stored identifier if the calculated distance is less than 20 pixels. If the distance exceeds 20 pixels, the algorithm identifies a new broiler that will receive a new identifier. This process is carried out for all the identified chicks in the current frame. Also, the maximum number of missed detections before the algorithm removes a tracking label is set to 10 frames.

### 2.4. Algorithm for Estimation of the Feeding Time

The feeding behavior of broiler chickens is related to a particular area where the broiler’s head is placed inside the feeder. In this study, there is a need to define the feeding area in the monitoring scene to investigate broilers’ feeding time. This area was manually determined in each video. In Figure 2, the blue circle represents the feeding area. The trained YOLO recognizes and marks the broiler head with a bounding box. Furthermore, the tracking algorithm tracks detected heads as long as they are inside the red area (Figure 2). When the center point of the head-labeled bounding box intersects and enters the feeding area (blue circle), feeding behavior may occur. The following index was used to determine whether a specific object covered a location (Equation (1)).
(1)$${Index}_{feed}=\frac{Area\;of\;head\cap Feeding\;area}{Feeding\;area}$$

An *Index_feed_* greater than 0 shows that the broiler has the head in the feeder. The feeding time of each broiler can be estimated by counting the number of frames that the broiler’s head is inside the feeding area. Accordingly, the feeding behavior can be detected along with the feeding time. Figure 3 and Figure 4 show the workflow and pseudocode of the developed algorithm for feeding time estimation.

The Python language was applied to write the algorithm used in the present research under the TensorFlow deep learning framework. Also, practical training was conducted using a computer with 32-core processors, 64 GB of RAM, and Nvidia Quadro RTX5000 16 GB graphics card.

## 3. Results

### 3.1. Head Detection

In the training step, 192,000 iterations were performed, and after every 100 iterations, the model weights were saved. Figure 5 illustrates the values of mean average precision (mAP) and loss function for model training. The error was very high in the first 200 iterations, fluctuated significantly, then decreased. This model did not have any significant performance increase after about 100,000 iterations. The highest performance was attained at iteration 132,996, in which the mAP and loss function values were 0.9320 and 0.0303, respectively. The trained model can recognize the broiler’s head when the image is captured. The detection result is determined by a bounding box that has a label and a number indicating the likelihood of belonging to this label. Figure 2 presents samples of the detection results.

### 3.2. Evaluating the Developed Algorithm

The developed algorithm can continuously judge feeding behavior occurrence in every frame. The algorithm was evaluated by selecting 3 1-min videos from 3 different days (consisting of 2280 frames), which were annotated manually and accurately. The labeling process consisted of two steps: (1) counting the number of broilers’ heads inside the feeding area (red area in Figure 2), to evaluate the algorithm’s performance in detecting and tracking heads, and (2) counting the number of frames in which each broiler pecks the feeding pan, to determine the actual value of the feeding time. The number of frames each broiler spends feeding can be obtained by applying the proposed algorithm to each video frame. Finally, by dividing the obtained number by the fps value, the feeder occupation time by the broiler’s head was calculated as feeding time.

Figure 6 presents the results of the diagnosis of feeding behavior for each video. For the first video, the number of detected heads was two more than the actual number. The same error can be seen in the other videos. This issue occurred due to an error in the detection model or tracking algorithm. It means that either the detection model failed to distinguish a specific head or the tracking algorithm assigned more than one identifier to one head. The developed algorithm achieved an overall head detection accuracy of 82.8%. The overall accuracy of the feeding time estimation and feeding time of each broiler per visit to the feeding pan was 97.9% and 87.3%, respectively. The developed algorithm demonstrated that each broiler spends a feeding time of 18.46 s per visit to the feeding pan. Some broilers occasionally spend a short feeding time. It is essential to mention that when the head of a specific broiler is inside the feeder, the feeding behavior may not occur, considered during the manual annotation of these videos. Distinguishing the exact feeding time when the broiler’s head is inside the feeder is a challenging image processing task. The mentioned point can explain the reason for less than 90% accuracy in estimating feeding time per broiler. The obtained results prove that the developed algorithm, as an automatic and non-invasive tool for feeding time estimation, can be utilized in commercial poultry farms for effective management.

## 4. Discussion

The study achieved an accuracy of 82.8% for broiler head detection and high accuracies of 97.9% and 87.3% for overall feeding time estimation and feeding time of each broiler per visit to the pan, respectively. Several studies have been conducted to investigate automated monitoring of animals’ feeding behavior. Li, Zhao, Purswell, Du, Chesser Jr, and Lowe [1] investigated group-reared broilers’ feeding and drinking behaviors with the image processing technique by determining the number of birds at the feeder and drinkers. The results demonstrated an 89–93% accuracy in determining the number of birds at the feeder. The limitation of this study was the inability of the proposed method to estimate the feeding time of individual broilers. Aydin and Berckmans [23] described a monitoring system to estimate the feeding behavior of broilers at the group level using pecking sound processing. They recorded the pecking sounds of 10 broilers with a microphone attached to the feeder. In their study, R^2^ = 0.965 was reported for meal size estimation. In spite of significant results, analyzing individual broilers’ sounds and feeding behavior was not performed in this study. In another effort, the Faster R-CNN model was employed to identify the feeding behavior of group-housed pigs [24]. In this study, the feeding behavior of four pigs on the experimental farm was investigated and the precision rate of 0.996 was obtained for feeding behavior recognition. Li et al. [25] monitored the feeding and drinking behaviors of broilers using an ultra-high-RFID-based system. The applied system included tags, antennas, a reader, and a data acquisition system. They investigated the feeding and drinking behaviors of 60 tagged broilers and achieved accuracies of 99.0% and 93.7% for determining time spent at the feeder and drinker, respectively. Despite satisfactory results, this system can monitor a limited number of broilers, and besides the sensor cost, installing the tags can result in stress and welfare issues. The Faster R-CNN model was also applied to recognize the number of pullets at each drinker and estimate the time spent at the drinker [26].

Table 1 summarizes some studies reporting automated monitoring of animals’ behavior. The data collected from a research farm or specific tools such as an ultra-high-RFID system, a compartment, a tag installed on broilers, or each nipple drinker were commonly utilized in these studies. In addition, they have researched group-housed poultry behaviors. In comparison, the highlights of this study include that (1) the proposed algorithm in this study was performed on a commercial farm; (2) the algorithm estimated the feeding time of each broiler per visit to the feeding pan without disrupting the regular farm operations.

However, it is worth mentioning that there are several limitations of this study and potential improvements that can be made in future studies. First, video data from this study were collected from only one commercial farm setting. Although the farm setting and the feeder pan are widely adopted among US broiler farms, more diverse data need to be used in order to conclude that our model is accurate for general commercial uses. In addition, training YOLO with a large image dataset collected from different farms can assist the model in generalizing well. Second, feeding area, defined as an area within a blue circle in Figure 2, was manually determined in this study. To make this algorithm fully automated, a simple feeder pan detection algorithm should be developed. Third, our algorithm considered a broiler as feeding when its head was in the feeding area. It is fine to define “feeding” behavior in this way; however, the method may not be accurate in estimating feed intake. In order to estimate true feed intake, the actual pecking behavior should be detected. Additional sensors, such as a microphone and pressure sensor, may be helpful for this purpose. Nevertheless, our study was one of the first efforts towards individual broiler feeding behavior analysis under commercial farm settings as mentioned in the highlights. We believe this study will be valuable as we continue moving towards individual broiler monitoring for precision management on commercial farms.

## 5. Conclusions

Understanding poultry feeding behavior is crucial for ensuring optimal growth, maintaining health and welfare, improving feed efficiency, managing economic implications, implementing appropriate nutritional management, and reducing the environmental impact of broiler production. Therefore, this study proposed an algorithm based on image processing techniques and a deep learning model to estimate broiler feeding time. Detection and tracking of the broiler’s head are essential in judging daily behaviors such as feeding behavior. Hence, the developed algorithm applied the YOLO-v3 model and Euclidean distance-based tracking algorithm to determine whether feeding behavior occurs for each surveillance video frame. Achieving state-of-the-art accuracy in the feeding time estimation of individual broilers can assist in developing efficient equipment for monitoring the broilers’ feeding behavior and providing valuable data for farm management. Our group continuously collects more data from various commercial farms to improve the accuracy of the developed algorithm and prepare the system for commercial farms globally.

## Figures and Tables

**Figure 1 animals-13-02428-f001:**
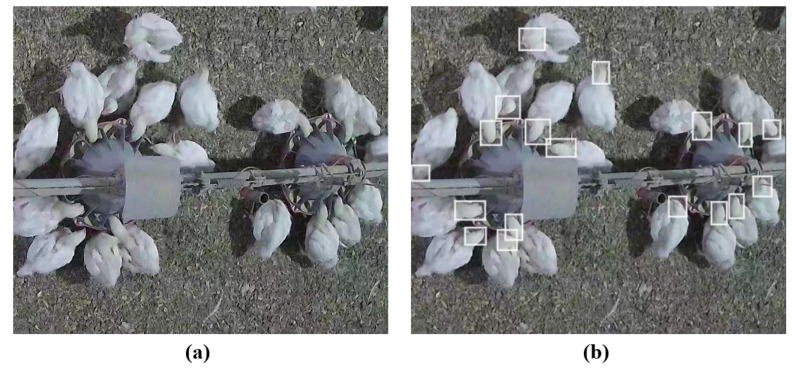
An instance of the labeling process for training the YOLO; (**a**) original image and (**b**) labeled image.

**Figure 2 animals-13-02428-f002:**
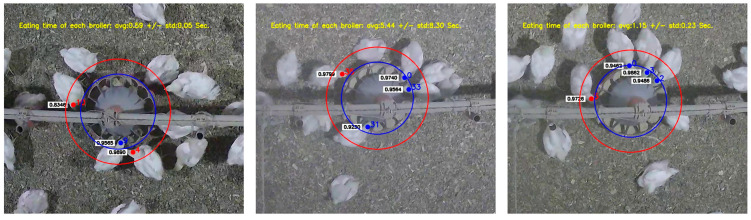
Samples of the developed algorithm’s result. Broilers’ heads with blue dots: feeding behavior; broilers’ heads with red dots: non-feeding behavior.

**Figure 3 animals-13-02428-f003:**
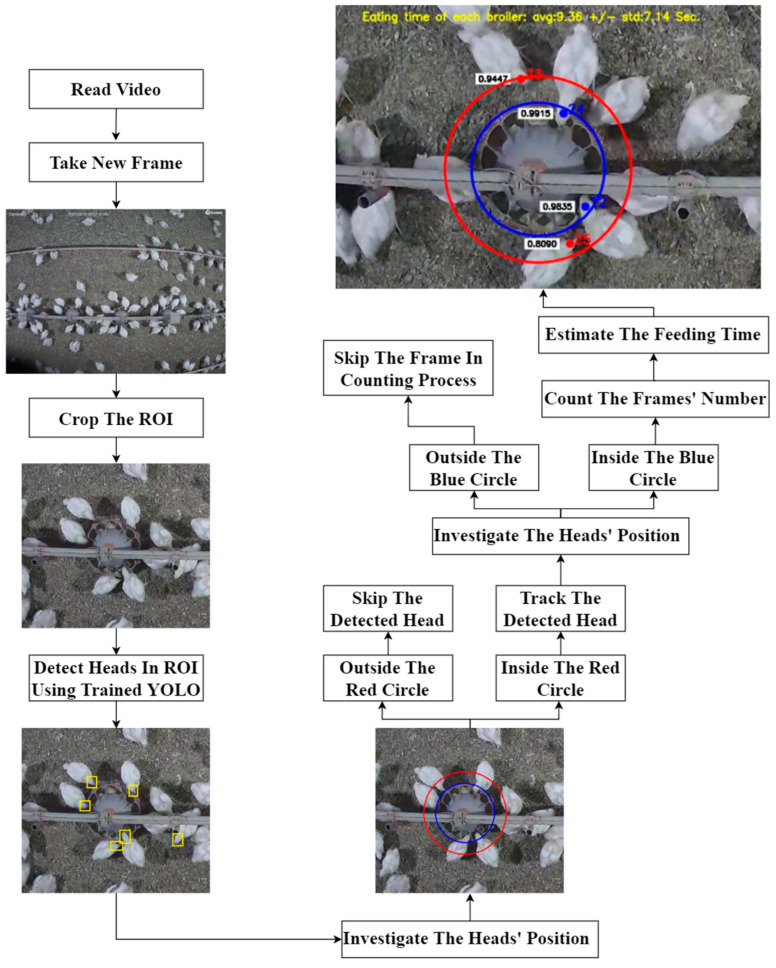
The workflow of the developed algorithm for feeding time estimation.

**Figure 4 animals-13-02428-f004:**
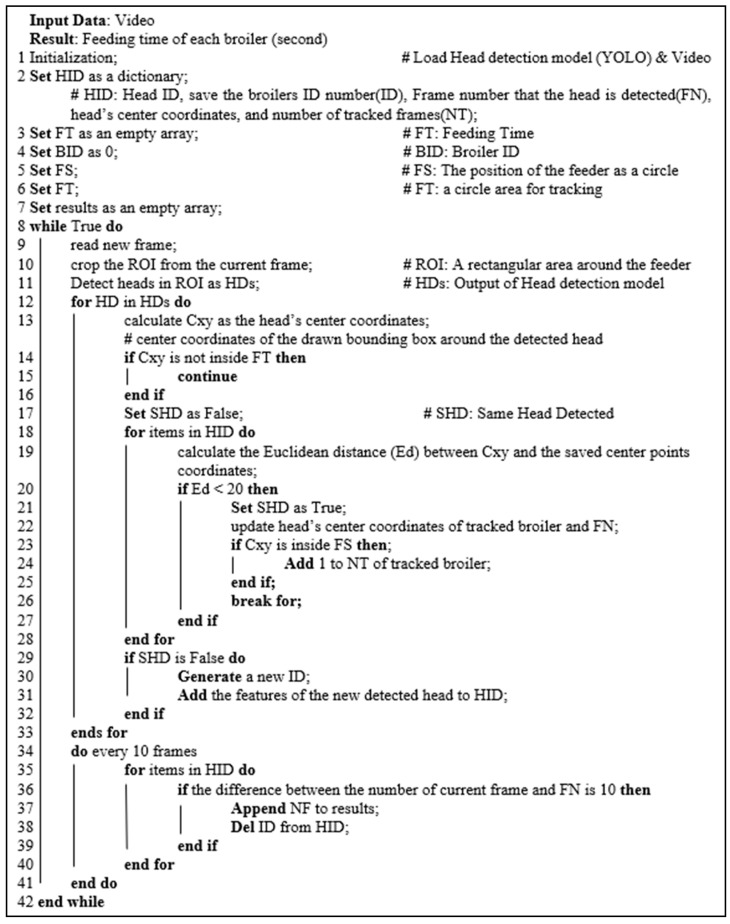
Pseudocode of the developed algorithm for feeding time estimation. “#” shows the comments.

**Figure 5 animals-13-02428-f005:**
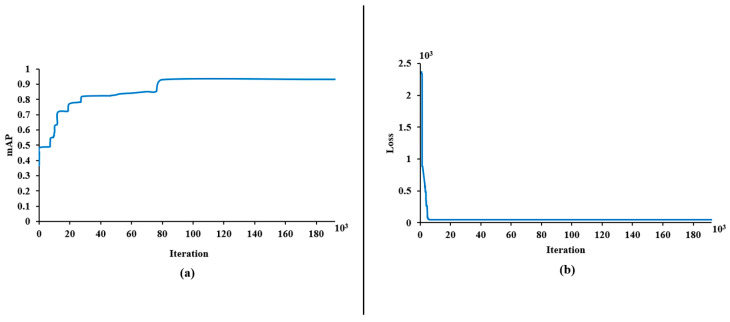
(**a**) Mean average precision (mAP) and (**b**) loss function for each iteration of the training process.

**Figure 6 animals-13-02428-f006:**
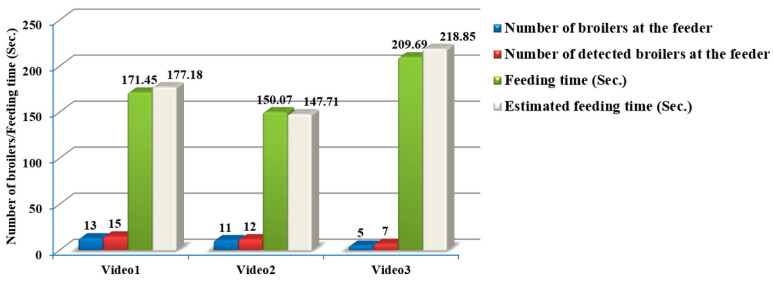
Results of evaluating the developed algorithm for three different 1-min videos.

**Table 1 animals-13-02428-t001:** Comparison of animals’ behavior monitoring accuracy of different studies and the present study.

Study	Equipment	Dataset	Class	Main Processing Method	Accuracy (%)
Li, Zhao, Purswell, Du, Chesser Jr, and Lowe [1]	RGB camera	Experimental farm	Group broilers	Image processing/Linear model	89–93
Aydin and Berckmans [23]	Microphone	Experimental farm	Group broilers	Sound analysis/Linear model	0.965
Yang, Xiao, and Lin [24]	RGB camera	Experimental farm	Individual pigs	Image processing/Machine learning	0.996
Li, Zhao, Hailey, Zhang, Liang, and Purswell [25]	Ultra-high-RFID	Experimental farm	Group broilers	Statistical analysis	93.7 and 99.0
Current study	RGB camera	Commercial farm	Individual broilers	Image processing/Machine learning	87.3 and 97.9

## Data Availability

The data that support the findings of this study are available on request from the corresponding author.

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
