# Peer review of "Estimating the Feeding Time of Individual Broilers via Convolutional Neural Network and Image Processing"

_animals, 2023, doi:10.3390/ani13152428_

Round 1

Reviewer 1 Report

This study's main contribution is the development and evaluation of an automated approach for estimating the feeding time of individual broilers in poultry farms. In summary, the research is interesting and provides valuable results, but the current document has several weaknesses that must be strengthened in order to obtain a documentary result that is equal to the value of the publication.

1. In the introduction, it is mentioned that machine learning methods can enhance the performance of image processing techniques. To support this claim, the author should include additional related references. For example, Valletta et al. utilized PCA for extracting features from wild eggs and employed k-means clustering to identify individual wild eggs. The inclusion of more relevant references will strengthen the argument.

2. It would be beneficial for the author to clarify whether the surveillance camera installed on the farm ceiling, as described in section 2.1, is fixed or capable of rotation. Additionally, it should be explained whether the 12 surveillance probes can capture RGB video footage of all 20,000 broiler chickens. Providing these details will help readers better understand the experimental setup.

3. The dataset was randomly divided into training and test sets at a ratio of 9:1. However, it is worth considering whether the addition of a validation set would be necessary. Including a validation set allows for better model performance assessment and hyperparameter tuning. The author should discuss the reasoning behind the choice of not including a validation set, if applicable.

4. It is suggested to include an original image in Figure 1 to provide readers with a visual reference. The inclusion of an original image will help readers better comprehend the subsequent image processing or analysis steps.

5. Vision technology integrated with deep learning is emerging these years in various engineering fields. The authors may add more state-of-art visual application articles for the integrity of the introduction (Detection and Counting of Banana Bunches by Integrating Deep Learning and Classic Image-Processing Algorithms; Computers and Electronics in Agriculture. Novel visual crack width measurement based on backbone double-scale features for improved detection automation; Engineering Structures. Identification and Detection of Biological Information on Tiny Biological Targets Based on Subtle Differences; Machines. ).

6. While the advantages of this paper are discussed in the Discussion section, it would be beneficial to compare the proposed method with other existing methods. Providing comparisons and highlighting the improvements achieved by this paper compared to other methods will enhance the clarity of the advantages presented.

7. To facilitate comparison with other methods, the author can consider including a table or a graph that showcases the performance of the proposed method alongside other relevant methods. This will provide readers with a comprehensive and visually accessible overview of the performance comparison.

8. In the Discussion section, it is important to analyze and discuss the reasons for the failure of certain tests or experiments. By identifying and explaining the potential causes of these failures, readers will gain a deeper understanding of the limitations or challenges faced by the proposed method. Therefore, the author should incorporate a thoughtful analysis of the reasons behind any failed tests within the Discussion.

Reviewer 2 Report

Feeding behavior is a crucial indicator of broiler welfare, providing valuable insights into resource utilization and farm management. Traditional monitoring methods relying on human observation are time-consuming and impractical for large-scale poultry farms. This study aims to address this challenge by developing an automated approach to estimate the feeding time of individual broilers using a convolutional neural network (CNN) and image processing techniques. The You Only Look Once (YOLO) model was trained on 1500 labeled images collected from a poultry farm to detect the broilers' heads. A Euclidean distance-based tracking algorithm was implemented to track the detected heads, enabling the estimation of feeding time by determining whether the broiler's head is inside the feeder. To evaluate the algorithm's performance, three 1-minute labeled videos were utilized. The results demonstrated an overall accuracy of 87.3% in estimating the feeding time of each broiler per visit to the feeding pan. The findings indicate that the proposed algorithm holds promise as a real-time tool for monitoring feeding behavior in poultry farms.

Feeding behavior plays a pivotal role in assessing the welfare of broilers, offering valuable insights into farm management and resource utilization. However, manual observation of feeding behavior is labor-intensive and time-consuming, making it impractical for large poultry farms. To overcome these challenges, this study proposes an automated approach using a convolutional neural network (CNN) and image processing techniques to estimate the feeding time of individual broilers. By leveraging computer vision technologies, this research aims to enhance efficiency and accuracy in monitoring poultry behaviors, providing a real-time tool for poultry farm management.

To evaluate the proposed algorithm's performance, three 1-minute videos with labeled ground truth data were employed. The algorithm achieved an overall accuracy of 87.3% in estimating the feeding time of individual broilers per visit to the feeding pan. These results demonstrate the effectiveness of the developed algorithm in accurately estimating feeding behavior in a poultry farm setting. The findings suggest that the proposed algorithm can be a valuable real-time tool for monitoring and managing broiler feeding behavior in large-scale poultry farms.

This study presents a novel approach for estimating the feeding time of individual broilers using a convolutional neural network and image processing techniques. The developed algorithm demonstrated high accuracy in estimating feeding behavior, providing a practical and efficient alternative to manual observation. By automating the monitoring process, this research contributes to improving welfare assessment and resource management in poultry farms. The proposed algorithm shows potential as a real-time tool for poultry farm management, enhancing productivity and ensuring optimal welfare conditions for broilers. Further research can explore the application of this algorithm in other poultry-related behaviors and expand its usability in the industry.

The quality of English language used in the provided text is generally good.

Reviewer 3 Report

I think this is a well-written article and an interesting study. 

Detailed comments: 

In the abstract and later on in the article it is mentioned that 1500 images were used to train and test the algorithm, and that 3 videos of 1 minute each were used to validate the model. Were labelled images from these videos used, and if so, how many images were labelled from the videos? Three videos of 1 minute seems not very much to me, especially since the variety of behaviors shown in this short time must be limited. I would prefer more videos (maybe of shorter timespans), for example more spread out over the day. Maybe eating behaviour varies during the day of over the weeks. 

L50 add 'it is

L58 from a hen - do you mean from each other? 

L233 several studies (not research) have been conducted

L234-241 in these lines, a few studies are mentioned but it is unclear how or in what sense, they are compared to the present study in L245-247. Are the highlights of this study, things that are different from the other studies? If so, why are they better? 

The fact that the study was performed on a commercial farm is mentioned first as a highlight, and than as a limitation. What is it now? 

L256 Why is it 'totally fine' to define eating behaviour as 'head in the feeding area'? Are there any validation studies in the references for that assumption? 

L265-271 these lines belong in the introduction, they are not conclusions. 

Reviewer 4 Report

Review

Article:

Estimating the Feeding Time of Individual Broilers via Convolutional Neural Network and Image Processing

 The research aimed to evaluate the feeding time of individual broilers with a convolutional neural network-based model.

Summary and Introduction:

The Summary states that the study aims to evaluate the feeding time of individual broilers. However, in the Introduction, the study's objective does not mention individual identification. This is important since individual identification is the genuine interest of commercial farms.

I believe more articles using YOLO for individual identification of animal behavioral patterns in the current literature should be included here. For instance:

Chandana, R. K., & Ramachandra, A. C. (2022). Real Time Object Detection System with YOLO and CNN Models: A Review. https://doi.org/10.48550/arXiv.2208.00773

Yılmaz, A., Uzun, G. N., Gürbüz, M. Z., & Kıvrak, O. (2021, August). Detection and breed classification of cattle using yolo v4 algorithm. In 2021 International Conference on INnovations in Intelligent SysTems and Applications (INISTA) (pp. 1-4). IEEE.

Amino, K., & Matsuo, T. (2022). Automated behavior analysis using a YOLO-based object detection system. In Behavioral Neurogenetics (pp. 257-275). New York, NY: Springer US.

Zhong, J., Li, M., Qin, J., Cui, Y., Yang, K., & Zhang, H. (2022). Real-time marine animal detection using YOLO-based deep learning networks in the coral reef ecosystem. The International Archives of the Photogrammetry, Remote Sensing and Spatial Information Sciences, 46, 301-306.

Schneider, S., Taylor, G. W., & Kremer, S. (2018, May). Deep learning object detection methods for ecological camera trap data. In 2018 15th Conference on Computer and robot vision (CRV) (pp. 321-328). IEEE.

Ma, D., & Yang, J. (2022, October). Yolo-animal: An efficient wildlife detection network based on improved yolov5. In 2022 International Conference on Image Processing, Computer Vision and Machine Learning (ICICML) (pp. 464-468). IEEE.

M & M

In general, the method is well-defined. However, it is unclear if the identification is of an individual bird. What if the same bird goes to the feeder, eats just a little portion, and returns? Can you identify the individual bird, as mentioned in the Summary? Does this identification really matter? Please check the manuscript in this direction.

Results

Lines 208-209: "Finally, the feeder occupation time is obtained for each 208 broiler, which is equivalent to the feeding time of the same broiler." This is not clear; please clarify this assumption.

Lines 219 -226: If you had difficulty processing the image to infer that the broiler head is on the feeder's top (or inside), how can you affirm that the model can be successfully used in a commercial farm?

Line 249: This should be emphasized in the M & M section. However, I believe that the number of samples rather than the number of farms should improve the accuracy.

 Line 260: It is not clear how the individual identification was done.

Review

Article:

Estimating the Feeding Time of Individual Broilers via Convolutional Neural Network and Image Processing

 The research aimed to evaluate the feeding time of individual broilers with a convolutional neural network-based model.

Summary and Introduction:

The Summary states that the study aims to evaluate the feeding time of individual broilers. However, in the Introduction, the study's objective does not mention individual identification. This is important since individual identification is the genuine interest of commercial farms.

I believe more articles using YOLO for individual identification of animal behavioral patterns in the current literature should be included here. For instance:

Chandana, R. K., & Ramachandra, A. C. (2022). Real Time Object Detection System with YOLO and CNN Models: A Review. https://doi.org/10.48550/arXiv.2208.00773

Yılmaz, A., Uzun, G. N., Gürbüz, M. Z., & Kıvrak, O. (2021, August). Detection and breed classification of cattle using yolo v4 algorithm. In 2021 International Conference on INnovations in Intelligent SysTems and Applications (INISTA) (pp. 1-4). IEEE.

Amino, K., & Matsuo, T. (2022). Automated behavior analysis using a YOLO-based object detection system. In Behavioral Neurogenetics (pp. 257-275). New York, NY: Springer US.

Zhong, J., Li, M., Qin, J., Cui, Y., Yang, K., & Zhang, H. (2022). Real-time marine animal detection using YOLO-based deep learning networks in the coral reef ecosystem. The International Archives of the Photogrammetry, Remote Sensing and Spatial Information Sciences, 46, 301-306.

Schneider, S., Taylor, G. W., & Kremer, S. (2018, May). Deep learning object detection methods for ecological camera trap data. In 2018 15th Conference on Computer and robot vision (CRV) (pp. 321-328). IEEE.

Ma, D., & Yang, J. (2022, October). Yolo-animal: An efficient wildlife detection network based on improved yolov5. In 2022 International Conference on Image Processing, Computer Vision and Machine Learning (ICICML) (pp. 464-468). IEEE.

M & M

In general, the method is well-defined. However, it is unclear if the identification is of an individual bird. What if the same bird goes to the feeder, eats just a little portion, and returns? Can you identify the individual bird, as mentioned in the Summary? Does this identification really matter? Please check the manuscript in this direction.

Results

Lines 208-209: "Finally, the feeder occupation time is obtained for each 208 broiler, which is equivalent to the feeding time of the same broiler." This is not clear; please clarify this assumption.

Lines 219 -226: If you had difficulty processing the image to infer that the broiler head is on the feeder's top (or inside), how can you affirm that the model can be successfully used in a commercial farm?

Line 249: This should be emphasized in the M & M section. However, I believe that the number of samples rather than the number of farms should improve the accuracy.

 Line 260: It is not clear how the individual identification was done.
